# Efficient Gradient Clipping Methods in DP-SGD for Convolution Models

## Abstract

Differentially private stochastic gradient descent (DP-SGD) is a well-known method for training machine learning models with a specified level of privacy. However, its basic implementation is generally bottlenecked by the computation of the gradient norm (gradient clipping) for each example in an input batch. While various techniques have been developed to mitigate this issue, there are only a handful of methods pertaining to convolution models, e.g., vision models. In this work, we present three practical methods for performing gradient clipping that improve upon previous state-of-art methods. Two of these methods use in-place operations to reduce memory overhead, while the third one leverages a relationship between Fourier transforms and convolution layers. We then develop a dynamic algorithm that dispatches one of the above three algorithms to optimize performance. Extensive benchmarks confirm that this algorithm consistently outperforms other state-of-the-art algorithms and frameworks.

## 1 Introduction

Differentially-private stochastic gradient descent (DP-SGD) is a common tool used to train machine learning models to protect sensitive information contained within individual training records (Abadi et al., 2016). However, general implementations of DP-SGD are bottlenecked by their gradient clipping step, whose runtime and memory costs scale linearly with the batch size times the number of model parameters. Our goal in this work is to develop three improved variants of the gradient clipping step that are substantially more efficient when applied to models with convolution layers.

*DP-SGD details*. The DP-SGD algorithm (Chaudhuri et al., 2011; Bassily et al., 2014) relies on the Gaussian mechanism and composition of differential privacy (Dwork et al., 2006; 2014) across iterations to privately compute the average of per-example gradients in a batch. At each iteration it operates by (i) bounding the *sensitivity* of each record within a batch to control and quantify the impact of any single record on the final model weights, and (ii) adding Gaussian noise proportional to the inverse of the batch size times the bound in (i). In particular, sensitivity is controlled by bounding per-example gradient norms so that the privatized gradients lie in a compact set. This approach is crucial for reducing noise growth, which scales as $\mathcal{O}(\sqrt{d}/[\epsilon b])$ (Bassily et al., 2014), where $d$ is the number of model parameters, $\epsilon$ the privacy budget, and $b$ the number of records in a batch. Alternatively, one can clip the overall average gradient at each step, but this increases noise by a factor of the batch size to $\mathcal{O}(\sqrt{d}/\epsilon)$.

Naive per-example clipping requires computing the norm of all per-example gradients. Specifically, this methods requires storing at least a matrix of size $\Theta(bd)$ that contains per-example gradients. Given the importance of model utility within this privacy-preserving context, there have been several developments on improving this step (with a focus on models with fully-connected or embedding layers). For example, techniques like ghost-clipping (Goodfellow, 2015) have been leveraged to improve both the runtime and storage complexity in certain settings. However, similar savings for convolution layers remain elusive (Rochette et al., 2019; Lee and Kifer, 2021a).

*Contributions*. This work introduces a meta-algorithm for gradient clipping for convolutional networks, which improves overall efficiency by selecting from three specialized methods, each outperforming prior techniques within distinct hyperparameter regimes. More specifically:

- the first two methods use in-place calculations and obtain $O(1)$ per-example storage complexities;
- the first method directly computes the squared norm, while the second leverages the ghost-clipping trick for fully-connected layers;
- the third method uses a relationship between convolution operators and fast Fourier transforms (FFTs) to obtain a scheme that scales well in the high-dimensional setting.

It is worth mentioning that the analysis for the third method (for gradient clipping) appears to be new. In particular, this analysis exploits properties of circulant matrices to derive an algorithm that runs efficiently in terms of the number of model parameters $d$, and the batch size $b$. Further, the efficiency of this method is most pronounced for large kernels that are designed to capture long-range dependencies, a setting required by several practical applications. As a byproduct, the proposed FFT approach also accelerates the computation of full CNN gradients in certain regimes, a benefit beyond the primary focus of our work.

To verify the practical efficiency of our methods, we provide benchmark experiments that demonstrate the numerical performance of our proposed methods outperforming popular frameworks.

*Related work.* The vast literature on DP-SGD (Chaudhuri et al., 2011; Bassily et al., 2014; Abadi et al., 2016; Ponomareva et al., 2023; Bu et al., 2023a) highlights the challenge of bounding individual record sensitivity, a crucial aspect often addressed through clipping[1]. While alternative approaches exist, such as modifying model architectures to enable Lipschitz constant computation (Béthune et al., 2023), their broader applicability remains uncertain.

To the best of our knowledge, the state-of-the-art performance in the setting of convolution models is achieved by Bu et al. (2023b). Specifically, that work builds on the approach of Bu et al. (2022); Lee and Kifer (2021a) and combines it with a careful book-keeping scheme that avoids a second back-propagation step. The main observation of Bu et al. (2022) is that the straightforward implementation of DP-SGD can be faster or more memory-efficient than ghost-clipping in certain regimes. While Rochette et al. (2019); Lee and Kifer (2021a) rely on instantiating per-example gradients, Bu et al. (2022) take advantage of the underlying network structure and choose which of of two different approaches to run; this selection step drives the bulk of their speed-up. In a follow-up work (Bu et al., 2023b), the authors use the previous observation and the idea that the second back-propagation step can be avoided using caching techniques.

Fourier transforms have first been used to improve the efficiency of training convolution neural networks (CNNs) by Mathieu et al. (2013), who build upon related work by Ben-Yacoub et al. (1999) for small-scale fully-connected models. Additional improvements to the approach have been developed, for example, by Pratt et al. (2017); Vasilache et al. (2014); Abtahi et al. (2017); Rippel et al. (2015). However, our development of similar techniques for the purpose of gradient clipping appears to be new.

For convenience, we compare in Table 1.1 the asymptotic time runtime and storage complexities of the ghost-clipping and direct methods by Bu et al. (2022); Lee and Kifer (2021a) and our proposed methods. Note that we only consider the ghost-clipping method of Bu et al. (2022) and **not** the mixed ghost-clipping method in (Bu et al., 2022, Algorithm 1) as the latter has complexity equal to the minimum of the former and the direct method of Lee and Kifer (2021a).

*Notation.* For a matrices $A$ and $B$ we let $\|A\|$ denote the Frobenius norm of $A$ and $\langle A, B \rangle$ denote the (Frobenius) inner product. Let $(\mathcal{W}, \langle \cdot, \cdot \rangle)$ and $(\mathcal{Y}, \langle \cdot, \cdot \rangle)$ denote two Hilbert spaces with common induced norm $\| \cdot \|$. We denote linear operators between them by italicized letters $\mathcal{A} \colon \mathcal{W} \to \mathcal{Y}$ and denote $\mathcal{A}^* \colon \mathcal{Y} \to \mathcal{W}$ to be the *adjoint* of $\mathcal{A}$. That is, $\mathcal{A}^*$ is the unique linear operator that satisfies

$$\langle y, \mathcal{A}w \rangle = \langle \mathcal{A}^* y, w \rangle \quad \forall w \in \mathcal{W}, \quad \forall y \in \mathcal{Y}. \tag{1}$$

Let $\psi \colon \mathcal{W} \to \mathcal{Y}$ be an arbitrary function. The *Fréchet derivative* of $\psi$ at $w_0 \in \mathcal{W}$ is given by the unique bounded linear operator $D\psi(w_0) \colon \mathcal{W} \to \mathcal{Y}$ satisfying

$$\lim_{\delta \to 0} \frac{\|\psi(w_0 + \delta) - \psi(w_0) - D\psi(w_0)\delta\|}{\|\delta\|} = 0.$$

---

[1]See Pichapati et al. (2019); Chen et al. (2020) for examples or Ponomareva et al. (2023) for a recent overview.

Table 1.1: Asymptotic time/space complexities of gradient clipping methods for a single example. The scalars $n_{\text{in}}$, $n_{\text{out}}$, $d_k$, $d_{\text{in}}$, and $d_{\text{out}}$ denote the number of input channels, output channels, kernel size, input dimension, and output dimension, respectively. Direct methods materialize the unaltered gradients, ghost-clipping methods apply the trick from Goodfellow (2015), and FFT methods utilize a novel relationship between convolution layers and FFTs proposed in this work.

| Method | Type | Runtime | Storage |
|---|---|---|---|
| Lee and Kifer (2021b) | direct | $n_{\text{in}}n_{\text{out}}d_{\text{out}}d_k$ | $n_{\text{out}}d_{\text{out}} + n_{\text{in}}d_{\text{out}}d_k$ |
| Bu et al. (2022) | ghost-clipping | $d_{\text{out}}^2(n_{\text{in}}d_k + n_{\text{out}})$ | $d_{\text{out}}^2 + n_{\text{out}}d_{\text{out}} + n_{\text{in}}d_{\text{in}}d_k$ |
| Algorithm 3.1 [ours] | direct | $n_{\text{in}}n_{\text{out}}d_{\text{out}}d_k$ | $O(1)$ |
| Algorithm 3.2 [ours] | ghost-clipping | $d_{\text{out}}^2(n_{\text{in}}d_k + n_{\text{out}})$ | $O(1)$ |
| Algorithm 3.3 [ours] | FFT | $n_{\text{in}}n_{\text{out}}d_{\text{in}}\log(d_{\text{in}})$ | $d_{\text{in}}$ |

We say $\psi$ is differentiable if its Fréchet derivative exists for all $w_0 \in \mathcal{W}$. Throughout this paper we will use two special properties of the Fréchet derivative: the chain rule and the existence of gradients. Let $(\mathcal{Z}, \langle \cdot, \cdot \rangle)$ be another Hilbert space and $\phi \colon \mathcal{Y} \to \mathcal{Z}$ be given. The *chain rule* provides us with a simple way to calculate the derivative of the function $\phi \circ \psi \colon \mathcal{W} \to \mathcal{Z}$, namely,

$$D(\phi \circ \psi)(w_0) = D\phi(\psi(w_0))D\psi(w_0)\,.$$

The Fréchet derivative of $\psi$ at $w_0$ with respect to a subset of variables $u$ is denoted by $D_u\phi(w_0)$. Finally, $\nabla\psi(w_0) \in \mathcal{W}$ denotes the (unique) gradient of a function $\psi$ at $w_0$, which satisfies

$$D\psi(w_0)\delta = \langle \nabla\psi(w_0), \delta \rangle_{\mathcal{W}} \quad \forall \delta \in \mathcal{W}\,. \tag{2}$$

The existence of the gradient is guaranteed by the well-known Riesz-Fréchet Representation Theorem (Rudin et al., 1976). The gradient of $\psi$ at $w_0$ with respect to a set of variables $u$ is denoted by $\nabla_u\psi(w_0)$.

*Organization.* Section 2 presents some necessary background material on representing gradient norms in convolution models. Section 3 presents the proposed clipping methods and discusses their properties and algorithm complexities under different regimes. Finally, Section 4 gives several numerical experiments and benchmarks.

## 2 BACKGROUND

To simplify our presentation, we focus on a single convolution layer and a single example $x \in \mathbb{R}^{n_{\text{in}} \times d_{\text{in}}}$ from the batch of inputs. For the case of multiple convolution layers and multiple examples, it is straightforward to see that our complexity results scale linearly with the number of layers times the number of examples. For conciseness, we present our results for one-dimensional inputs, but discuss the generalization to high dimensions in Section 3.

Given a stride length $s \geq 1$, let $d_k \in \mathbb{N}$, $d_{\text{in}} \in \mathbb{N}$, $d_{\text{out}} = 1 + (d_{\text{in}} - d_k)/s$ be the size[2] of the kernel, inputs, and outputs, respectively, let $n_{\text{in}} \in \mathbb{N}$ and $n_{\text{out}} \in \mathbb{N}$ be the number of input, output channels, respectively, and let $w \in \mathbb{R}^{n_{\text{in}} \times n_{\text{out}} \times d_k}$ be the kernel weights. Moreover, for fixed output channel $j$, let (i) $w^{i,j} \in \mathbb{R}^{d_k}$ be the kernel vector corresponding to the $i$-th input channel, (ii) $b^j \in \mathbb{R}^{n_{\text{out}} \times d_{\text{out}}}$ be the bias offset, (iii) $\alpha$ be a general activation function, and (iv) $U_x^i \in \mathbb{R}^{d_{\text{out}} \times d_k}$ be a matrix whose $\ell$-th row consists of the entries in the $i$-th input channel of $x$ that are being multiplied with $w^{i,j}$.

The output for the $j$-th output channel of a convolution layer is given by

$$[\phi_x(w,b)]^j = \phi_x^j(w,b) := \alpha \left( b^j + \sum_{i \in [n_{\text{in}}]} U_x^i w^{i,j} \right)\,. \tag{3}$$

---

[2]To avoid clutter, we assume these are all integers. In the implementation of our approach, we handle the general case.

Numerically efficient schemes for computing $\|\nabla_b \phi_x^j(w,b)\|^2$ (the bias weights' gradient norm), have been previously developed by Kong and Munoz Medina (2023). Consequently, our focus is on analyzing the kernel weights' gradient norm $\|\nabla_w \phi_x^j(w,b)\|^2$. Following similar analyses as Kong and Munoz Medina (2023), we first write

$$\phi_x^j = \ell_x \circ \psi_x^j \circ Z_x \quad \text{where} \quad \psi_x^j(z) := \alpha(z + b^j), \quad Z_x^j(w) := \sum_{i \in [n_{\text{in}}]} U_x^i w^{i,j}. \tag{4}$$

Then, if we denote

$$\mathcal{A} = \mathcal{A}_x(w) := DZ_x^j(w), \qquad g^j = g_x^j(w) := \nabla(\ell_x \circ \psi_x^j)(Z_x^j(w)), \tag{5}$$

it follows from the chain rule that

$$\|\nabla_w \phi_x^j(w,b)\|^2 = \Omega_x(g^j) := \left\|\mathcal{A}^* g^j\right\|^2. \tag{6}$$

To avoid the notational clutter, we denote the adjoint operator of $DZ_x^j(w)$ by $DZ_x^{j*}(w)$. Using the fact that $\nabla_w \phi_x(w,b) = [\nabla_w \phi_x^1(w,b), \dots, \nabla_w \phi_x^{n_{\text{out}}}(w,b)]$, we have that

$$\|\nabla_w \phi_x(w,b)\|^2 = \sum_{j=1}^{n_{\text{out}}} \|\nabla_w \phi_x^j(w,b)\|^2 = \sum_{j=1}^{n_{\text{out}}} \left\|\mathcal{A}^* g^j\right\|^2,$$

and, hence, it suffices to restrict our presentation to a fixed output channel $j$ where applicable. Kong and Munoz Medina (2023) established efficient representations of $\Omega_x(g)$ for the case of embedding and fully-connected layers. Similarly, our task will be to find an efficient representation of $\Omega_x(g)$ for convolution layers.

## 3 Algorithms and discussion

This section comprises three subsections detailing the main sub-algorithms and technical aspects of our work. The first subsection introduces the in-place algorithms and their characteristics. The second discusses the Fourier-based algorithm and its properties. The final subsection compares these methods across various regimes, considering factors like input-output channels and dimensions. This analysis carefully characterizes the optimality regimes for each sub-algorithm, yielding the meta-algorithm that selects the most suitable method for a given layer.

Before proceeding, we describe some common notation and a basic result about the function $Z_x(\cdot)$ in (5). Given a 4D array $M \in \mathbb{R}^{n_{\text{in}} \times n_{\text{out}} \times d_k \times d_{\text{out}}}$, we denote $M_{m,\ell}^{i,j}$ to be the value in the corresponding to the $i$-th input channel, $j$-th output channel, $m$-th input dimension, and $\ell$-th output dimension of $M$. We give similar definitions for the arrays/scalars $M^{i,j}$, $M_m^i$, $M_\ell^j$, $M^i$, and $M^j$, keeping the convention that superscripts (resp. subscripts) contain indices for the input/output channels (resp. dimensions). The straightforward representation of the operators we have discussed so far requires defining and handling fourth-order tensors, which can vastly complicate the analysis. However, we are able to decompose various operations across different channels and dimensions, which allows us to only use two-dimensional matrices to represent all the operators we use.

The result below provides some convenient representations of the Fréchet derivative of $Z_x^j(w)$ and $Z_x^{j*}(w)$. Its proof is postponed to Appendix A.

**Lemma 3.1.** *Let $U_x^i \in \mathbb{R}^{d_{\text{out}} \times d_k}$ be as in (4) for some input channel $i \in [n_{\text{in}}]$, let $\Delta \in \mathbb{R}^{n_{\text{in}} \times n_{\text{out}} \times d_k}$, and $\tau^j \in \mathbb{R}^{d_{\text{out}}}$ be arbitrary. If $\Delta^{i,j} \in \mathbb{R}^{d_k}$ is the displacement vector corresponding to input-output channel pair $(i,j) \in [n_{\text{in}}] \times [n_{\text{out}}]$, then*

(a) $DZ_x^j(w)[\Delta] = \sum_{i \in [n_{\text{in}}]} U_x^i \Delta^{i,j} \in \mathbb{R}^{d_{\text{out}}}$;

(b) $\{DZ_x^{j*}(w)[\tau^j]\}^{i,j} = [U_x^i]^* \tau^j \in \mathbb{R}^{d_k}$;

(c) $DZ_x^j(w) \circ DZ_x^{j*}(w)[\tau^j] = \sum_{i \in [n_{\text{in}}]} U_x^i [U_x^i]^* \tau^j \in \mathbb{R}^{d_{\text{out}}}$.

Since the elements of $U_x^i$ are the values of $x$, the identity in (6) and Lemma 3.1(b) imply that the squared norm of $\nabla_w \phi_x^j(w,b)$ can be expressed solely in terms of $x$ and the downstream gradient $g^j$ in (5). In the next two subsections, we give two different expressions for $\|\nabla_w \phi_x^j(w,b)\|$ and present their corresponding algorithms.

## 3.1 Memory-efficient norm computation

This subsection presents two in-place algorithms for computing the desired squared gradient norm.

We first present a "direct" expression for $\nabla_w \phi_x^j(w, b)$ in terms of $x$ and $g^j$ using (6). The proof is postponed to Appendix A.

**Lemma 3.2.** *Let $g^j \in \mathbb{R}^{n_{\text{out}} \times d_{\text{out}}}$ be as in (6) and $s \geq 1$ be given. Then, it holds that the value of the gradient $\nabla_w \phi_x^j(w, b)$ at the $i$-th input channel, $j$-th output channel, and $m$-th output dimension is given by*

$$[\nabla_w \phi_x^j(w, b)]_m^{i,j} = \sum_{\ell \in [d_{\text{out}}]} (x_{[\ell-1]s+m}^i)(g_\ell^j). \tag{7}$$

The above result shows that when we are given $x$ and $g$, we can compute $\|\nabla_w \phi_x(w, b)\|^2$ by performing a sequence of in-place operations. For ease of reference, we present one variant of these operations in Algorithm 3.1, which can be viewed as an in-place modification of the FastGradClip algorithm in Lee and Kifer (2021b). It is immediate that Algorithm 3.1 requires

$$T_{\text{direct}} := n_{\text{in}} n_{\text{out}} d_k d_{\text{out}} \tag{8}$$

floating-point operations (FLOPS), but only $O(1)$ additional storage.

---

**Algorithm 3.1** Direct squared norm computation with in-place operations

---

1: *Input*: stride length $s \geq 1$, layer input $x \in \mathbb{R}^{n_{\text{in}} \times d_{\text{in}}}$, and gradient $g \in \mathbb{R}^{n_{\text{out}} \times d_{\text{out}}}$;
2: *Output*: value of $\|\nabla_w \phi_x(w, b)\|^2$;
3: Define $\mathcal{J}_m := \{([\ell-1]s + m, \ell) : \ell \in [d_{\text{out}}]\}$ for $m \in [d_k]$

4: **return** $\sum_{i \in [n_{\text{in}}]} \sum_{j \in [n_{\text{out}}]} \sum_{m \in [d_k]} \left( \sum_{(p,q) \in \mathcal{J}_m} x_p^i g_q^j \right)^2$

---

We now present a special expression for $\|\nabla_w \phi_x(w, b)\|^2$ that is reminiscent of a similar expression in the "Ghost Clipping" algorithm from Bu et al. (2022). The proof is postponed to Appendix A.

**Lemma 3.3.** *Let $g^j \in \mathbb{R}^{d_{\text{out}}}$ be as in (6), let $s \geq 1$ be given, and define*

$$X_{\ell,\ell'} := \sum_{i \in [n_{\text{in}}]} \sum_{m \in [d_k]} (x_{[\ell-1]s+m}^i)(x_{[\ell'-1]s+m}^i), \quad G_{\ell,\ell'} := \sum_{j \in [n_{\text{out}}]} g_\ell^j g_{\ell'}^j$$

*where $\ell, \ell' \in [d_{\text{out}}]$ are indices over the output dimension. Then, it holds that*

$$\|\nabla_w \phi_x(w, b)\|^2 = \sum_{j \in [n_{\text{out}}]} \left\langle \mathcal{A}_x \mathcal{A}_x^*, [g^j][g^j]^* \right\rangle = 2 \sum_{1 \leq \ell < \ell' \leq d_{\text{out}}} X_{\ell,\ell'} G_{\ell,\ell'} + \sum_{\ell \in [d_{\text{out}}]} X_{\ell,\ell} G_{\ell,\ell}, \tag{9}$$

*where $\mathcal{A}_x$ is the matrix in $\mathbb{R}^{d_{\text{out}} \times d_k}$ corresponding the operator of the same name in (5).*

Similar to Lemma 3.2, the above result also yields a sequence of in-place operations for computing $\|\nabla_w \phi_x(w, b)\|^2$. As before, for ease of reference, we present one variant of these operations in Algorithm 3.2. It is straightforward to see that, for a fixed outer index pair $(\ell, \ell')$ in the expression for $P$, the computation of the inner sum involving $x$ (resp. $g$) requires $n_{\text{in}} d_k$ FLOPS (resp. $n_{\text{out}}$). Consequently, computing $P$ and $Q$ in Algorithm 3.2 requires

$$T_{\text{ghost}} := \left[ d_{\text{out}} + \frac{d_{\text{out}}(d_{\text{out}} - 1)}{2} \right] (n_{\text{in}} d_k + n_{\text{out}}) = \Theta(d_{\text{out}}^2 [n_{\text{in}} d_k + n_{\text{out}}]) \tag{10}$$

total FLOPS but also only $O(1)$ additional storage.

---

**Algorithm 3.2** Ghost Clipping-based squared norm computation with in-place operations

---

1: *Input*: layer input $x \in \mathbb{R}^{n_{\text{in}} \times d_{\text{in}}}$ and gradient $g \in \mathbb{R}^{n_{\text{out}} \times d_{\text{out}}}$;
2: *Output*: value of $\|\nabla_w \phi_x(w, b)\|^2$;
3: Define $\mathcal{J}_{\ell,\ell'} := \{([\ell-1]s + m, [\ell'-1]s + m) : m \in [d_k]\}$ for $\ell, \ell' \in [d_{\text{out}}]$
4: Compute $P \leftarrow \sum_{1 \leq \ell < \ell' \leq d_{\text{out}}} \left( \sum_{i \in [n_{\text{in}}]} \sum_{(p,q) \in \mathcal{J}_{\ell,\ell'}} x_p^i x_q^i \right) \left( \sum_{j \in [n_{\text{out}}]} g_\ell^j g_{\ell'}^j \right)$

5: Compute $Q \leftarrow \sum_{\ell \in [d_{\text{out}}]} \left( \sum_{i \in [n_{\text{in}}]} \sum_{(p,q) \in \mathcal{J}_{\ell,\ell}} x_p^i x_q^i \right) \left( \sum_{j \in [n_{\text{out}}]} g_\ell^j g_\ell^j \right)$
6: **return** $2P + Q$

---

## 3.2 FOURIER-BASED NORM COMPUTATION

This subsection presents an algorithm based on the discrete Fourier transform (DFT) for computing the desired squared gradient norm.

We first define $\mathrm{rev} : \mathbb{R}^n \mapsto \mathbb{R}^n$ (resp. $\mathrm{diag} : \mathbb{R}^n \mapsto \mathbb{R}^{n \times n}$) to be the linear operator that reverses the order of its input (resp. diagonalizes its input). Explicitly, these operators are given by

$$\mathrm{rev}([x_1, x_2, \ldots, x_n]) = [x_n, \ldots, x_2, x_1], \quad [\mathrm{diag}(x)]_{i,j} = \begin{cases} x_i, & \text{if } i = j \\ 0, & \text{otherwise} \end{cases}, \quad \forall i, j \in [n], \quad (11)$$

for every $x \in \mathbb{R}^n$. Now, let us recall the notion of a circulant matrix and its relationship to the DFT. A circulant matrix $C \in \mathbb{R}^{n \times n}$ (resp. an anti-circulant matrix $\zeta \in \mathbb{R}^{n \times n}$) is a Toeplitz (resp. anti-Toeplitz) matrix of the form

$$C = \begin{bmatrix} c_0 & c_{n-1} & \cdots & c_1 \\ c_1 & c_0 & \cdots & c_2 \\ \vdots & \vdots & \ddots & \vdots \\ c_{n-1} & c_{n-2} & \cdots & c_0 \end{bmatrix}, \quad \zeta = \begin{bmatrix} c_1 & \cdots & c_{n-1} & c_0 \\ c_2 & \cdots & c_0 & c_1 \\ \vdots & \cdot\cdot\cdot & \vdots & \vdots \\ c_0 & \cdots & c_{n-2} & c_{n-1} \end{bmatrix}, \quad (12)$$

for some $c \in \mathbb{R}^n$. Notice that consecutive rows of a circulant (resp. anti-circulant) matrix contain the same entries of $c$ but are cyclically shifted from left to right (resp. right to left).

The next result relates circulant matrices in $\mathbb{R}^{n \times n}$ with the $n$-th order DFT, and its proof can be found, for example, in (Gray et al., 2006).

**Lemma 3.4.** *If $C \in \mathbb{R}^{n \times n}$ is a circulant matrix and $c$ is its first column, then $C = \mathcal{F}_n^{-1} \mathrm{diag}(\mathcal{F}_n c) \mathcal{F}_n$, where $\mathcal{F}_n$ is the $n$-th order DFT.*

Using the above result, it is straightforward to see that if $\zeta \in \mathbb{R}^{n \times n}$ is an anti-circulant matrix whose first row is $\mathrm{rev}(c)$, then

$$\zeta \tau = \mathrm{rev}(\mathcal{F}_n^{-1} \mathrm{diag}[\mathcal{F}_n \mathrm{rev}(c)] \mathcal{F}_n \tau), \quad \forall \tau \in \mathbb{R}^n. \quad (13)$$

Returning to our main goal, the primary insight of this section is that we can express $\nabla_w \phi_x^j(w, b)$ (and, consequently, $\nabla_w \phi_x(w, b)$) as an application of an anti-circulant matrix with simple linear transforms. The details of this perspective, and its computational implications, are given in the following result, whose proof is postponed to Appendix A.

**Proposition 3.5.** *Let $\zeta_x^i \in \mathbb{R}^{d_{\mathrm{in}} \times d_{\mathrm{in}}}$ denote the anti-circulant matrix whose first row is $x^i$. Moreover, define the block matrices $Q \in \mathbb{R}^{d_{\mathrm{in}} \times d_k}$ and $R \in \mathbb{R}^{d_{\mathrm{out}} \times d_{\mathrm{in}}}$ by*

$$Q := \begin{bmatrix} I_{d_k} \\ 0_{(d_{\mathrm{in}} - d_k) \times d_k} \end{bmatrix}, \quad [R]_{n,m} = \begin{cases} 1, & \text{if } m = s(n-1) + 1 \\ 0, & \text{otherwise} \end{cases} \quad \forall (n, m) \in [d_{\mathrm{in}}] \times [d_{\mathrm{out}}], \quad (14)$$

*where $I_n$ (resp. $0_{n \times m}$) denotes the identity matrix in $\mathbb{R}^{n \times n}$ (resp. zero matrix in $\mathbb{R}^{n \times m}$). Then, it holds that*

*(a) for every $i \in [n_{\mathrm{in}}]$, we have $U_x^i = R \zeta_x^i Q$;*

*(b) if $g^j \in \mathbb{R}^{d_{\mathrm{out}}}$ is as in (5), then*

$$\left[ \nabla_w \phi_x^j(w, b) \right]^i = Q^* \circ \mathrm{rev} \circ \mathcal{F}_{d_{\mathrm{in}}}^{-1} \left( [\mathcal{F}_{d_{\mathrm{in}}} \circ \mathrm{rev}(x^i)] \odot [\mathcal{F}_{d_{\mathrm{in}}} R^* g^j] \right) \quad \forall i \in [n_{\mathrm{in}}], \quad (15)$$

*where $\odot$ denotes the Hadamard product.*

Before proceeding, let us give a few remarks. First, for $y \in \mathbb{R}^{d_{\mathrm{in}}}$ and $z \in \mathbb{R}^{d_{\mathrm{out}}}$, we have that $Q^* y$ returns the first $d_k$ rows of $y$ and $R^* z$ returns a padded version of $z$ in which $[R^* z]_{s(i-1)+1} = z_i$ for $i \in [d_{\mathrm{out}}]$ and $[R^* z]_j$ is zero at all other indices $j$. Second, in view of the first remark, we have that for any $y \in \mathbb{R}^{d_{\mathrm{in}}}$, both of the quantities $(Q^* \circ \mathrm{rev})(y)$ and $R^* g^j$ can be computed using $d_k$ and $d_{\mathrm{out}}$ FLOPS, respectively.

We now present a general algorithm in Algorithm 3.3 that leverages (15) to calculate $\|\nabla_w \phi_x(w, b)\|^2$. Notice, in particular, that it can be specialized to different choices of the DFT oracle $\mathcal{F}_{d_{\text{in}}}$.

---

**Algorithm 3.3** DFT-based squared norm computation

---

1: *Input*: layer input $x \in \mathbb{R}^{n_{\text{in}} \times d_{\text{in}}}$, gradient $g \in \mathbb{R}^{n_{\text{out}} \times d_{\text{out}}}$, and oracle $\mathcal{F}_{d_{\text{in}}}$ that performs the $(d_{\text{in}})$-th order DFT;
2: *Output*: value of $\|\nabla_w \phi_x(w, b)\|^2$;
3: Define $\text{rev}(\cdot)$ and $(Q, R)$ to be as in (11) and (14), respectively
4: **for** $i, j \in [n_{\text{in}}] \times [n_{\text{out}}]$ **do**
5: $\quad v^{i,j} \leftarrow Q^* \circ \text{rev} \circ \mathcal{F}_{d_{\text{in}}}^{-1}([\mathcal{F}_{d_{\text{in}}} \circ \text{rev}(x^i)] \odot [\mathcal{F}_{d_{\text{in}}} R^* g^j])$
6: $\quad r^{i,j} \leftarrow \sum_{\ell=1}^{d_k} (v_\ell^{i,j})^2$
7: **end for**
8: **return** $\sum_{(i,j) \in [n_{\text{in}}] \times [n_{\text{out}}]} r^{i,j}$

---

The next result presents the runtime and storage complexity of a specialization of Algorithm 3.3, where we use a fast discrete Fourier transform (DFT) oracle. Specifically, it is well-known that DFT can be implemented in $O(d \log d)$ time complexity (Duhamel and Vetterli, 1990). The proof can be found in Appendix A.

**Theorem 3.6.** *Let $\bar{\mathcal{F}}_{d_{\text{in}}}$ be an FFT oracle which, for any $v \in \mathbb{R}^{d_{\text{in}}}$, computes $\bar{\mathcal{F}}_{d_{\text{in}}} v$ and $\bar{\mathcal{F}}_{d_{\text{in}}}^{-1} u$ in $T_{\mathcal{F}} = \Theta(d_{\text{in}} \log d_{\text{in}})$ FLOPS. Then, there is an implementation of Algorithm 3.3 with $\mathcal{F}_{d_{\text{in}}} = \bar{\mathcal{F}}_{d_{\text{in}}}$ that consumes at most*

$$T_{\text{fft}} := n_{\text{in}} n_{\text{out}} (d_{\text{out}} + d_{\text{in}} + 3d_k + 3T_{\mathcal{F}}) = \Theta(n_{\text{in}} n_{\text{out}} d_{\text{in}} \log d_{\text{in}}) \tag{16}$$

*total FLOPS and $\Theta(d_{\text{in}})$ additional storage.*

Let us now compare the runtime complexities $T_{\text{direct}}$, $T_{\text{ghost}}$, and $T_{\text{fft}}$ in (8), (10), and (16), respectively. For simplicity, let us assume that the stride length is $s = 1$ and let $d \geq 1$ and $n \geq 1$ be arbitrary. Denoting $A \preceq B$ to mean $A$ is asymptotically more efficient than $B$ in terms of runtime, we observe the relationships for different choices of $n_{\text{in}}$, $n_{\text{out}}$, $d_k$, $d_{\text{in}}$, and $d_{\text{out}}$ in Table 3.1.

Table 3.1: Relationships of asymptotic runtimes across key parameter regimes.

| $d_{\text{in}}, d_{\text{out}}$ | $d_k$ | $n_{\text{in}}, n_{\text{out}}$ | Relationships |
|---|---|---|---|
| $\Theta(d)$ | $\Theta(d)$ | $O(1)$ | $T_{\text{fft}} = \Theta(d \log d) \preceq T_{\text{direct}} = \Theta(d^2) \preceq T_{\text{ghost}} = \Theta(d^3)$ |
| $\Theta(d)$ | $\Theta(1)$ | $O(1)$ | $T_{\text{direct}} = \Theta(d) \preceq T_{\text{fft}} = \Theta(d \log d) \preceq T_{\text{ghost}} = \Theta(d^2)$ |
| $\Theta(d)$ | $\Theta(d)$ | $O(n)$ | $T_{\text{ghost}} = \Theta(nd) \preceq T_{\text{direct}} = \Theta(n^2 d) \preceq T_{\text{fft}} = \Theta(n^2 d \log d)$ |

In particular, we mention that the large kernel regime in the first row of Table 3.1 was recently studied in Ding et al. (2022) and was show to have significant utility improvements compared to the small kernel regime of the second row.

**Optimal subroutine selection**. Table 3.1 shows that no clipping method is universally superior, and that the optimal choice depends the layer's specific input-output channels $(n_{\text{in}}, n_{\text{out}})$ and dimensions $(d_k, d_{\text{in}}, d_{\text{out}})$. In view of this fact, we propose a simple meta-algorithm in Algorithm 3.4 that optimally (in terms of runtime) dispatches the best of our proposed subroutines (Algorithms 3.1– 3.3) for a given set of parameters. Clearly, the runtime of Algorithm 3.4 is $\min(T_{\text{direct}}, T_{\text{ghost}}, T_{\text{fft}})$.

---

**Algorithm 3.4** Meta-algorithm for gradient norm computation

---

1: *Input*: stride length $s \geq 1$, layer input $x \in \mathbb{R}^{n_{\text{in}} \times d_{\text{in}}}$, gradient $g \in \mathbb{R}^{n_{\text{out}} \times d_{\text{out}}}$, and oracle $\mathcal{F}_{d_{\text{in}}}$ that performs the $(d_{\text{in}})$-th order DFT in $T_{\mathcal{F}}$ FLOPS;
2: *Output*: value of $\|\nabla_w \phi_x(w, b)\|^2$;
3: Define $T_{\text{direct}}$, $T_{\text{ghost}}$, and $T_{\text{fft}}$ in (8), (10), and (16), respectively
4: Define $A_{\text{direct}}$, $A_{\text{ghost}}$, and $A_{\text{fft}}$ to be Algorithms 3.1–3.3, respectively
5: Compute $\pi = \text{argmin}\{T_p : p \in \{\text{direct}, \text{ghost}, \text{fft}\}\}$
6: Run algorithm $A_\pi$ with the appropriate inputs and return its output

---

## 3.3 Higher dimensions

While our results are formally presented for one-dimensional inputs with multiple channels, we conclude by discussing their generalization to higher-dimensional settings. When $x$ is an $d$-dimensional input, it is straightforward to develop analogous version for Algorithms 3.1–3.2. However, the analogous version of Algorithm 3.3 requires more care. In particular, we would need to develop higher-order versions of (12), replace the one-dimensional Fourier transform in Algorithm 3.3 with its $d$-dimensional variant, and replace the operators $(Q, R)$ in Algorithm 3.3 with higher-order variants.

In the special case of the two-dimensional DFT, used in the vast majority of modern deep learning architectures, e.g. when the inputs are images, it is known (Azimi-Sadjadi and King, 1987) that a version of Lemma 3.4 holds where $C$ is replaced by a block-circulant matrix, i.e., where each $c_i$ in (12) is replaced by a matrix. Consequently, the version of Algorithm 3.3 for a two-dimensional (per-example) input array $x$ directly follows from this result by replacing (i) $\mathcal{F}_{d_{\mathrm{in}}}$ by its analogous two-dimensional DFT, (ii) $\mathrm{rev}(\cdot)$ by the operator that reverses a two-dimensional input array lexicographically, and (iii) $Q$ and $R$ by their block two-dimensional variants. We posit that the $d$-dimensional version of Algorithm 3.3 is one where changes (i)–(iii) are applied in the $d$-dimensional setting, i.e., with blocks of $d$-dimensional arrays instead of two-dimensional matrices. For a more precise treatment of the two-dimensional case, see Appendix D.

## 4 Numerical experiments

As Algorithms 3.1 and 3.2 are primarily memory-efficient variants of the corresponding methods in Bu et al. (2022) and Lee and Kifer (2021b), we focus on benchmarking our FFT-based method (Algorithm 3.3), which uses a completely new technique. More specifically, we consider the large-kernel parameter regime in the first row of Table 3.1, where our FFT-based method has a distinct advantage in terms of the hyperparameter $d$.

All experiments were run in Python on an Ubuntu 22.04 instance with an Intel Xeon 2.20 GHz CPU, an NVIDIA Telsa T4 GPU with 15GB of VRAM, and 13.6GB of RAM.

It is also worth mentioning that we did not benchmark our FFT method with the large kernel models in Ding et al. (2022) because those models employ 4D kernels, while our paper primarily considers 1D kernels. While we present the generalization of Algorithm 3.3 to 2D kernels in Appendix D, we believe that the $n$D kernel requires significantly more development in both theory (e.g., algorithmic complexity) and implementation (e.g., custom FFT and sparse matrix GPU kernels). Consequently, we believe such experiments/benchmarks to be out-of-scope.

### 4.1 Gradient norm computation on CPU and GPU.

Tables 4.1–4.2 respectively present CPU and GPU runtimes and memory measurements of compute the gradient norm of a single one-dimensional convolution layer with $d_{\mathrm{in}} = d$, $d_k = d/2$, $n = 3$, and a single example as input. Due to the improved parallelism of GPUs, larger values of $d$ appear in Table 4.2 compared to the values in Table 4.1.

Table 4.1: Gradient norm runtime (ms) and peak RAM (MB) measurements on CPU for large kernel sizes. Values are rounded to the nearest whole number and are the median over five trials.

| | $d$ | 0.1k | 0.2k | 0.4k | 0.8k | 1.6k | 3.2k | 6.4k | 12.8k | 25.6k |
|---|---|---|---|---|---|---|---|---|---|---|
| **Runtime** | Algorithm 3.3 | 1 | 1 | 1 | 1 | 2 | 2 | 4 | 7 | 10 |
| | Lee and Kifer (2021b) | 0 | 0 | 1 | 4 | 14 | 69 | 279 | 1092 | 24852 |
| | Bu et al. (2022) | 1 | 1 | 2 | 7 | 28 | 185 | 1120 | 7788 | 78773 |
| **RAM** | Algorithm 3.3 | 0 | 0 | 0 | 0 | 0 | 0 | 1 | 1 | 2 |
| | Lee and Kifer (2021b) | 0 | 0 | 1 | 5 | 21 | 82 | 328 | 1311 | 5243 |
| | Bu et al. (2022) | 0 | 0 | 1 | 5 | 21 | 82 | 328 | 1311 | 5243 |

Table 4.2: Gradient norm runtime (ms) and change in VRAM (MB) measurements on GPU for large kernel sizes. Values are rounded to the nearest whole number and are the median over five trials. A value of "-" indicates that the GPU ran out of VRAM before completing a trial.

| | $d$ | 4k | 8k | 16k | 32k | 64k | 128k | 256k | 512k | 1024k |
|---|---|---|---|---|---|---|---|---|---|---|
| Runtime | Algorithm 3.3 | 2 | 2 | 2 | 3 | 3 | 3 | 3 | 7 | 13 |
| | Lee and Kifer (2021b) | 2 | 3 | 9 | 34 | 456 | - | - | - | - |
| | Bu et al. (2022) | 20 | 102 | 963 | 8625 | - | - | - | - | - |
| VRAM | Algorithm 3.3 | 0 | 2 | 4 | 10 | 19 | 36 | 63 | 124 | 245 |
| | Lee and Kifer (2021b) | 48 | 193 | 770 | 3072 | 12294 | - | - | - | - |
| | Bu et al. (2022) | 99 | 388 | 1544 | 6149 | - | - | - | - | - |

We now make a few remarks about the trends in Tables 4.1–4.2. First, consistent with the theory of Section 3, Algorithm 3.3 scales the most efficiently in terms of CPU/GPU runtime/memory usage, followed by the algorithm in Lee and Kifer (2021b), and then the one in Bu et al. (2022). Second, the RAM usage of the competing methods were nearly identical across trials while the VRAM used by the method in Bu et al. (2022) scaled worse than the one in Lee and Kifer (2021b). Finally, we note that on GPUs, our FFT algorithm (Algorithm 3.3) was able to evaluate gradient norms whose layer input dimension $d_{\mathrm{in}}$ was at least 16x larger than the best competing method (without going out-of-memory in VRAM).

## 4.2 END-TO-END TRAINING ON GPU

Table 4.3 presents runtimes for end-to-end training of a one-layer one-dimension convolution neural network with mean squared error in Opacus. More specifically, for each measurement, we ran one trial of five iterations of DP-SGD on GPU with batch size 128, $d_{\mathrm{in}} = d$, $d_k = d/2$, $n = 1$, $s = 1$, and either Opacus' implementation of DP-SGD (Naive DP-SGD), which materializes gradients directly, or the ghost norm variant of DP-SGD where the norm computation uses our FFT method (Algorithm 3.3 + DP-SGD). For more details on the Opacus implementation, see Appendix B.

This comparison highlights that gradient clipping imposes a non-trivial bottleneck in practical training frameworks like Opacus, an overhead that can be mistakenly overlooked compared to other fixed costs. The performance gains from our custom kernel (Algorithm 3.3) highlight the importance of optimizing this specific step.

Table 4.3: End-to-end runtime (s) measurements on GPU. Values are rounded to the nearest tenth for a single trial. A value of "-" indicates that the GPU ran out of VRAM before completing the trial.

| $d$ | 0.5k | 1k | 2k | 4k | 8k | 16k | 32k | 64k |
|---|---|---|---|---|---|---|---|---|
| Algorithm 3.3 + DP-SGD | 0.6 | 0.5 | 2.1 | 11.2 | 41.3 | 146.4 | 546.0 | 2310.9 |
| Naive DP-SGD | 0.3 | 1.0 | 4.9 | 16.8 | 72.8 | - | - | - |

From the results in Table 4.3, we can see that Naive DP-SGD performs well for smaller values of $d$ but scales significantly worse in $d$. Moreover, we obtained results with Algorithm 3.3 + DP-SGD on problem instances at least 8x larger than Naive DP-SGD (without going out-of-memory in VRAM). Note that Naive DP-SGD employs a single forward/backward pass, whereas Algorithm 3.3 + DP-SGD employs one forward pass and two backward passes.

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

## A    TECHNICAL PROOFS

This appendix gives the proofs of the manuscript's main results.

*Proof of Lemma 3.1.* For simplicity, denote $U^i = U_x^i$ and $\mathcal{D}_x := DZ_x(w)$.

(a) This is immediate from the linearity of $\mathcal{D}_x$.

(b) From part (a) and the definition of the adjoint, we have

$$\langle \mathcal{D}_x \Delta, \tau \rangle = \sum_{i \in [n_{in}]} \langle \tau, U^i \Delta^{i,j} \rangle = \sum_{i \in [n_{in}]} \langle [U^i]^* \tau, \Delta^{i,j} \rangle = \langle \mathcal{D}_x^* \tau, \Delta \rangle .$$

(c) Using parts (a) and (b), we have $\mathcal{D}_x \mathcal{D}_x^* \tau = \sum_{i \in [n_{in}]} U^i [\mathcal{D}_x^* \tau]^{i,j} = \sum_{i \in [n_{in}]} U^i [U^i]^* \tau.$ $\quad\square$

*Proof of Lemma 3.2.* In view of Lemma 3.1(b) and (6) it suffices to show that the $m$-th column of $U_x^i$ is the vector $\mathrm{Col}_m^i := [x_m^i, x_{s+m}^i, \ldots, x_{(d_{out}-1)s+m}^i] \in \mathbb{R}^{d_{out}}$. Indeed, recall that the $\ell$-th row of $U_x^i$ is the $\ell$-th window of the input $x$ and is given by $\mathrm{Row}_k^i := [x_{(\ell-1)s+1}^i, \ldots, x_{(\ell-1)s+d_k}^i] \in \mathbb{R}^{d_k}$. Fixing a column index $m$, it is clear that the values in the $m$-th index of $\mathrm{Col}_k^i$ for $k \in [d_k]$ form the elements of $\mathrm{Col}_m^i$. $\quad\square$

*Proof of Lemma 3.3.* The first identity in (9) is immediate from (6) and the definition of the adjoint of a linear operator. For the second identity, note that (4), (5), and Lemma 3.1(c) imply that $\mathcal{A}_x = \sum_{i \in [n_{in}]} U_x^i [U_x^i]^*$. Hence, in view of the definition of $X_{\ell,\ell'}$ and $G_{\ell,\ell'}$, it suffices to show that the entry in the $\ell$-th row and $\ell'$-th column of $U_x^i [U_x^i]^*$ is given by

$$\left[ U_x^i \{U_x^i\}^* \right]_{\ell,\ell'} = \sum_{m \in [d_k]} (x_{[\ell-1]s+m}^i)(x_{[\ell'-1]s+m}^i).$$

Indeed, recall that the $k$-th row of $U_x^i$, say $\mathrm{Row}_k^i$, contains the $\ell$-th window of the input array $x$. For a given stride $s$ and kernel size $d_k$, clearly we have $\mathrm{Row}_k^i = [x_{(\ell-1)s+1}^i, \ldots, x_{(\ell-1)s+d_k}^i].$ $\quad\square$

*Proof of Proposition 3.5.* (a) Observe that for any matrix $M \in \mathbb{R}^{d_{in} \times d_{in}}$, we have that $MQ$ returns the first $d_k$ columns of $M$ and $RM$ returns rows $1, s+1, \ldots, d_{out} - 1 + s$ of $M$. The conclusion now follows from the previous observation and the fact that the rows of $U_x^i$ contain the windows of $x$ of size $d_k$ and stride $s$.

(b) Using part (a) and (13) with $\zeta = \zeta_x^i$, we have that, for any $\tau \in \mathbb{R}^{d_k}$,

$$[U_x^i]^* \tau = Q^* [\zeta_x^i]^* R^* \tau = Q^* \circ \mathrm{rev}\left( \mathcal{F}_{d_{in}}^{-1} \mathrm{diag}\left[ \mathcal{F}_{d_{in}} \mathrm{rev}(x^i) \right] \mathcal{F}_{d_{in}} R^* \tau \right)$$

$$= Q^* \circ \mathrm{rev} \circ \mathcal{F}_{d_{in}}^{-1}\left( \left[ \mathcal{F}_{d_{in}} \mathrm{rev}(x^i) \right] \odot \left[ \mathcal{F}_{d_{in}} R^* \tau \right] \right)$$

Consequently, using the above identity with $\tau = g^j$ and Lemma 3.1(b) we have that

$$[\nabla_w \phi_x^j(w, b)]^i = [U_x^i]^* g^j = Q^* \circ \mathrm{rev} \circ \mathcal{F}_{d_{in}}^{-1}\left( \left[ \mathcal{F}_{d_{in}} \mathrm{rev}(x^i) \right] \odot \left[ \mathcal{F}_{d_{in}} R^* g^j \right] \right) .$$

$\quad\square$

*Proof of Theorem 3.6.* It suffices to describe the costs of computing $v^{i,j}$ and $r^{i,j}$ for $i \in [n_{in}]$ and $j \in [n_{out}]$.

For fixed $(i, j)$, computing $v^{i,j}$ can be done by: (i) computing $a = R^* g^j$ in a $d_{out}$ runtime and storage cost (see the remarks following Proposition 3.5), (ii) computing $\hat{a} = \mathcal{F}_{d_{in}} a$ and $\hat{c} = \mathcal{F}_{d_{in}} \circ \mathrm{rev}(x^i)$ in a $2T_{\mathcal{F}}$ runtime cost, (iii) computing $\hat{e} = \hat{c} \odot \hat{a}$ in a $d_{in}$ runtime and storage cost, (iv) computing $e = \mathcal{F}_{d_{in}}^{-1} \hat{e}$ in a $T_{\mathcal{F}}$ runtime cost, and (v) computing $Q^* \circ \mathrm{rev}(e)$ in a $d_k$ runtime and storage cost (see the remarks following Proposition 3.5). Summing the previous terms results in a

$$d_{out} + d_{in} + d_k + 3T_{\mathcal{F}}$$

runtime cost and fixed $\Theta(d_{\text{in}})$ storage cost. For fixed $(i, j)$, computing $r^{i,j}$, given $v^{i,j}$, can be done by an accumulating sum in a runtime and storage cost of $2d_k$ and $O(1)$, respectively.

Summing all the above costs over $i \in [n_{\text{in}}]$ and $j \in [n_{\text{out}}]$ (new temporary variables for the computations of $v^{i,j}$ and $r^{i,j}$) yields a storage cost of $\Theta(d_{\text{in}})$ and a runtime cost of

$$
n_{\text{in}}n_{\text{out}}\left(\underbrace{d_{\text{out}} + d_{\text{in}} + d_k + 3T_{\mathcal{F}}}_{v^{i,j}} + \underbrace{2d_k}_{r^{i,j}}\right) = T_{\text{fft}} = \Theta(n_{\text{in}}n_{\text{out}}[d_{\text{in}} \log d_{\text{in}}]),
$$

where the last identity follows from the fact that $d_k \leq d_{\text{in}}$. $\qquad\square$

## B    INTEGRATION OF ALGORITHM 3.3 IN OPACUS

This appendix gives a brief description of the integration of Algorithm 3.3 in the DP-SGD API of Opacus.

As Algorithm 3.3 is a algorithm for efficiently computing gradient norms, we create an Opacus-compatible norm sampler function that implements Algorithm 3.3 and register it with the global `GradSampleModuleFastGradientCLipping.NORM_SAMPLERS` attribute using Opacus' `@register_norm_sampler(...)` Python function decorator for `torch.nn.Conv1d` layers.

We then develop a benchmarking function using the tests inside the following Opacus module:

`opacus/tests/grad_sample_module_fast_gradient_clipping_test.py`

More specifically, this function explicitly performs the single forward and two backward passes employed by ghost norm variants of DP-SGD.

## C    ADDITIONAL NUMERICAL EXPERIMENTS AND GRAPHS

This appendix presents additional numerical experiments involve our proposed in-place algorithms, and the settings in this section are motivated by the three rows in Table 3.1.

Table C.1 presents runtime and peak RAM usage under the same setting as in Section 4 and corresponds to the first row of Table 3.1. Table C.2 considers the parameter setting $d_{\text{in}} = d$, $d_k = d - 13$, $n = 3$, and $s = 1$ and corresponds to the second row of Table 3.1. Finally, Table C.3 fixes $d_{\text{in}} = d_k = 10$ and $s = 1$, varies the number of channels $n$, and corresponds to the third row of Table 3.1. The choices of $n$ and $d$ in all tables are chosen to highlight the general trends of each algorithm compared to their competitors rather than push the limits of their computing environment.

Before proceeding, we mention that we were unable to implement runtime efficient variants of Algorithms 3.1–3.2 on GPU nor the same two algorithms within the parameter settings of rows 1 and 3 in Table 3.1, respectively. Consequently, we remove the above mentioned trials and focus on settings where our in-place algorithms perform well. On the other hand, we do note that our in-place implementations are still significantly more memory efficient than their counterparts in Bu et al. (2022); Lee and Kifer (2021b) on all the problem instances we tested.

We now make a few remarks about Tables C.1–C.3. First, we describe some results that are consistent with the theory in Section 3 (more specifically, Table 3.1): (a) the in-place algorithms (Algorithms 3.1–3.2) were significantly more efficient in peak RAM than their counterparts in Bu et al. (2022); Lee and Kifer (2021b), (b) Algorithm 3.2 is the most runtime efficient in Table C.2 followed by Algorithm 3.1, and (c) Algorithm 3.3 scales poorly when $n$ is increased in Table C.3. Second, we were unable to completely remove in the materialization of some small intermediate matrices in Algorithm 3.1, which causes some RAM to be consumed in the trials in Tables C.1–C.2.

## D    EXTENSION OF FFT ANALYSIS TO $2D$ CONVOLUTIONS

We now give the details of the generalization of the Fourier-based norm computation (Algorithm 3.3) to two-dimensional inputs, which is particularly relevant for vision models. The conceptual extension

Table C.1: Extra gradient norm runtime (ms) and peak RAM (KB) measurements on CPU for large kernel sizes. Values are rounded to the nearest whole number and are the median over five trials.

| | $d$ | 0.4k | 0.8k | 1.6k | 3.2k | 6.4k |
|---|---|---|---|---|---|---|
| Runtime | Algorithm 3.3 | 1 | 1 | 2 | 2 | 4 |
| | Algorithm 3.1 | 5 | 5 | 7 | 13 | 89 |
| | Lee and Kifer (2021b) | 1 | 4 | 14 | 78 | 292 |
| | Bu et al. (2022) | 2 | 6 | 29 | 194 | 1124 |
| RAM | Algorithm 3.3 | 34 | 68 | 135 | 270 | 539 |
| | Algorithm 3.1 | 164 | 646 | 2570 | 10256 | 40993 |
| | Lee and Kifer (2021b) | 1288 | 5134 | 20507 | 81974 | 327792 |
| | Bu et al. (2022) | 1288 | 5134 | 20507 | 81973 | 327789 |

Table C.2: Extra gradient norm runtime (ms) and peak RAM (KB) measurements on CPU for small kernel sizes. Values are rounded to the nearest whole number and are the median over five trials.

| | $d$ | 64k | 128k | 256k | 512k | 1024k |
|---|---|---|---|---|---|---|
| Runtime | Algorithm 3.3 | 47 | 60 | 145 | 328 | 802 |
| | Algorithm 3.1 | 30 | 24 | 44 | 209 | 279 |
| | Algorithm 3.2 | 44 | 54 | 63 | 115 | 180 |
| | Lee and Kifer (2021b) | 20 | 41 | 89 | 178 | 385 |
| | Bu et al. (2022) | 24 | 35 | 67 | 162 | 319 |
| RAM | Algorithm 3.3 | 5377 | 10753 | 21512 | 43020 | 86038 |
| | Algorithm 3.1 | 3842 | 7682 | 15363 | 30729 | 61453 |
| | Algorithm 3.2 | 1 | 4 | 3 | 3 | 8 |
| | Lee and Kifer (2021b) | 28668 | 57340 | 114684 | 229379 | 458758 |
| | Bu et al. (2022) | 28668 | 57340 | 114684 | 229379 | 458758 |

is straightforward, relying on the generalization of the Convolution Theorem and the properties of circulant matrices to higher dimensions.

*2D Notation.* We adapt the notation to accommodate two spatial dimensions. Let the input be $x \in \mathbb{R}^{n_{\text{in}} \times H_{\text{in}} \times W_{\text{in}}}$, where $H_{\text{in}}$ and $W_{\text{in}}$ are the input height and width, respectively. The kernel weights are $w \in \mathbb{R}^{n_{\text{in}} \times n_{\text{out}} \times H_k \times W_k}$, where $H_k$ and $W_k$ are the kernel height and width. We denote the stride lengths as $(s_H, s_W)$, resulting in output dimensions $H_{\text{out}}$ and $W_{\text{out}}$. The downstream gradient is $g \in \mathbb{R}^{n_{\text{out}} \times H_{\text{out}} \times W_{\text{out}}}$.

*Generalized Operators.* We must define the 2D counterparts of the operators introduced in Section 3.2.

- $\mathcal{F}_{2D}$ denotes the two-dimensional Discrete Fourier Transform.

- $\text{rev}_{2D}$ denotes the 2D reversal operator, which flips a 2D array across both spatial dimensions (horizontal and vertical), generalizing the $\text{rev}$ operator in (11).

- $Q_{2D}$ and $R_{2D}$ are the block two-dimensional variants of the operators defined in (14). Specifically, $R_{2D}^*$ maps the downstream gradient $g^j$ back to the input dimensions, accounting for the strides $(s_H, s_W)$. This corresponds to a dilation operation, inserting zeros between the elements of $g^j$. $Q_{2D}^*$ acts as a cropping operator, extracting the relevant $H_k \times W_k$ block corresponding to the kernel dimensions.

Table C.3: Extra gradient norm runtime (ms) and peak RAM (KB) measurements on CPU for different channel counts. Values are rounded to the nearest whole number and are the median over five trials.

| | $n$ | 40 | 80 | 160 | 320 | 640 |
|---|---|---|---|---|---|---|
| Runtime | Algorithm 3.3 | 110 | 298 | 1062 | 4766 | 19153 |
| | Algorithm 3.2 | 3 | 3 | 7 | 14 | 34 |
| | Lee and Kifer (2021b) | 1 | 1 | 2 | 5 | 16 |
| | Bu et al. (2022) | 1 | 1 | 1 | 3 | 5 |
| RAM | Algorithm 3.3 | 6 | 11 | 51 | 52 | 56 |
| | Algorithm 3.2 | 1 | 1 | 1 | 1 | 1 |
| | Lee and Kifer (2021b) | 100 | 294 | 1065 | 4144 | 16444 |
| | Bu et al. (2022) | 16 | 32 | 63 | 127 | 252 |

*2D Gradient Norm Computation.* The fundamental insight utilized in Proposition 3.5—that the convolution operator can be represented via circulant structures—generalizes to 2D. In this setting, 2D convolution is represented by Block Circulant with Circulant Blocks (BCCB) matrices. As noted in Section 3.3, a version of Lemma 3.4 holds where BCCB matrices are diagonalized by the 2D DFT (Azimi-Sadjadi and King, 1987). This allows us to derive the 2D equivalent of the identity presented in (15).

**Proposition D.1.** *(2D Extension of Proposition 3.5). Let $x^i \in \mathbb{R}^{H_{\text{in}} \times W_{\text{in}}}$ be the input for the $i$-th input channel, and let $g^j \in \mathbb{R}^{H_{\text{out}} \times W_{\text{out}}}$ be the downstream gradient for the $j$-th output channel. Let $Q_{2D}$ and $R_{2D}$ be the block two-dimensional operators defined above. Then, the gradient with respect to the kernel weights $w^{i,j}$ can be computed as:*

$$\left[\nabla_w \phi_x^j(w,b)\right]^i = Q_{2D}^* \circ \text{rev}_{2D} \circ \mathcal{F}_{2D}^{-1}\left([\mathcal{F}_{2D} \circ \text{rev}_{2D}(x^i)] \odot [\mathcal{F}_{2D} R_{2D}^* g^j]\right) \quad \forall i \in [n_{\text{in}}], \quad (17)$$

*where $\odot$ denotes the Hadamard product.*

*Proof.* We aim to derive the expression for the gradient $G^{i,j} := \left[\nabla_w \phi_x^j(w,b)\right]^i$ utilizing the 2D Discrete Fourier Transform ($\mathcal{F}_{2D}$).

**1. Gradient Expression as Cross-Correlation.** First, we observe the generalization of Lemma 3.2 to the 2D case. The value of the gradient with respect to the kernel weights $w^{i,j}$ at spatial location $(m,n) \in [H_k] \times [W_k]$ is given by:

$$G_{m,n}^{i,j} = \sum_{h=1}^{H_{\text{out}}} \sum_{w=1}^{W_{\text{out}}} (x_{[h-1]s_H+m,[w-1]s_W+n}^i) \cdot (g_{h,w}^j). \quad (18)$$

This operation represents a 2D cross-correlation between the input $x^i$ and the gradient $g^j$, accounting for the strides $(s_H, s_W)$.

**2. Incorporating Operators $R_{2D}$ and $Q_{2D}$.** We use the operators $R_{2D}^*$ and $Q_{2D}^*$ to manage the striding and dimensions explicitly. $R_{2D}^*$ represents the adjoint of the striding operation, which corresponds to dilation (zero-insertion). Let $\tilde{g}^j = R_{2D}^* g^j$. This maps $g^j$ into a sparse array aligned with the input spatial dimensions $D_{\text{in}}$.

$Q_{2D}^*$ represents the adjoint of the kernel embedding, which corresponds to cropping the result to the kernel dimensions $H_k \times W_k$.

The gradient computation in (18) can be conceptualized as the 'valid' cross-correlation between $x^i$ and the dilated gradient $\tilde{g}^j$, followed by the application of the cropping operator:

$$G^{i,j} = Q_{2D}^*(x^i \star \tilde{g}^j),$$

where $\star$ denotes 2D cross-correlation.

**3. Relating Cross-Correlation to Convolution.** To leverage the efficiency of the FFT via the Convolution Theorem, we must express the cross-correlation in terms of convolution. We utilize the identity involving spatial reversal ($\mathrm{rev}_{2D}$):

$$A \star B = \mathrm{rev}_{2D}((\mathrm{rev}_{2D}(A)) * B),$$

where $*$ denotes 2D convolution. Applying this to our gradient expression:

$$x^i \star \tilde{g}^j = \mathrm{rev}_{2D}((\mathrm{rev}_{2D}(x^i)) * \tilde{g}^j).$$

**4. Applying the 2D Convolution Theorem.** Let $\tilde{x}^i = \mathrm{rev}_{2D}(x^i)$. We now consider the convolution $C = \tilde{x}^i * \tilde{g}^j$. Assuming appropriate padding to ensure the DFT computes the required linear convolution (which is standard practice when implementing FFT-based convolutions), the 2D Convolution Theorem states:

$$C = \mathcal{F}_{2D}^{-1}(\mathcal{F}_{2D}(\tilde{x}^i) \odot \mathcal{F}_{2D}(\tilde{g}^j)),$$

where $\odot$ is the Hadamard product.

**5. Final Expression.** Substituting these identities back into the expression for $G^{i,j}$:

$$
\begin{aligned}
G^{i,j} &= Q_{2D}^* \circ \mathrm{rev}_{2D}(C) \\
&= Q_{2D}^* \circ \mathrm{rev}_{2D} \circ \mathcal{F}_{2D}^{-1}(\mathcal{F}_{2D}(\tilde{x}^i) \odot \mathcal{F}_{2D}(\tilde{g}^j)).
\end{aligned}
$$

Substituting the definitions $\tilde{x}^i = \mathrm{rev}_{2D}(x^i)$ and $\tilde{g}^j = R_{2D}^* g^j$, we arrive at the desired result:

$$\left[\nabla_w \phi_x^j(w,b)\right]^i = Q_{2D}^* \circ \mathrm{rev}_{2D} \circ \mathcal{F}_{2D}^{-1}\left([\mathcal{F}_{2D} \circ \mathrm{rev}_{2D}(x^i)] \odot [\mathcal{F}_{2D} R_{2D}^* g^j]\right).$$

$\square$

Proposition D.1 directly yields the 2D generalization of Algorithm 3.3.

---

**Algorithm D.1** 2D DFT-based squared norm computation

---

1: *Input*: layer input $x \in \mathbb{R}^{n_{\mathrm{in}} \times H_{\mathrm{in}} \times W_{\mathrm{in}}}$, gradient $g \in \mathbb{R}^{n_{\mathrm{out}} \times H_{\mathrm{out}} \times W_{\mathrm{out}}}$, and oracle $\mathcal{F}_{2D}$ that performs the 2D DFT;
2: *Output*: value of $\|\nabla_w \phi_x(w,b)\|^2$;
3: Define $\mathrm{rev}_{2D}(\cdot)$ and the block operators $(Q_{2D}, R_{2D})$.
4: **for** $i, j \in [n_{\mathrm{in}}] \times [n_{\mathrm{out}}]$ **do**
5: $\quad \hat{x}^i \leftarrow \mathcal{F}_{2D} \circ \mathrm{rev}_{2D}(x^i)$
6: $\quad \hat{g}^j \leftarrow \mathcal{F}_{2D} R_{2D}^* g^j$
7: $\quad v^{i,j} \leftarrow Q_{2D}^* \circ \mathrm{rev}_{2D} \circ \mathcal{F}_{2D}^{-1}(\hat{x}^i \odot \hat{g}^j)$
8: $\quad r^{i,j} \leftarrow \|v^{i,j}\|^2$ (Frobenius norm squared)
9: **end for**
10: **return** $\sum_{(i,j) \in [n_{\mathrm{in}}] \times [n_{\mathrm{out}}]} r^{i,j}$

---

*Complexity Analysis.* We analyze the complexity of Algorithm D.1 by extending Theorem 3.6. Let $D_{\mathrm{in}} = H_{\mathrm{in}} W_{\mathrm{in}}$ be the total spatial dimension of the input per channel. The complexity of the 2D FFT on an $H_{\mathrm{in}} \times W_{\mathrm{in}}$ input is $T_{\mathcal{F}, \in \mathcal{D}} = \Theta(D_{\mathrm{in}} \log D_{\mathrm{in}})$.

**Theorem D.2.** *(2D Extension of Theorem 3.6). Let $\bar{\mathcal{F}}_{2D}$ be a 2D FFT oracle. There is an implementation of Algorithm D.1 that consumes at most*

$$T_{\mathrm{fft,2D}} = \Theta(n_{\mathrm{in}} n_{\mathrm{out}} D_{\mathrm{in}} \log D_{\mathrm{in}}) = \Theta(n_{\mathrm{in}} n_{\mathrm{out}} H_{\mathrm{in}} W_{\mathrm{in}} \log(H_{\mathrm{in}} W_{\mathrm{in}})) \tag{19}$$

*total FLOPS and $\Theta(D_{\mathrm{in}})$ additional storage.*

The proof follows the same logic as the proof of Theorem 3.6, as the algorithm structure remains the same and the dominant cost is the execution of the 2D FFTs for each input-output channel pair. This confirms that the asymptotic advantages observed in the 1D case, particularly in the large kernel regime (Table 3.1), carry over directly to the 2D setting.

*Proof.* We analyze the computational complexity of Algorithm D.1. We assume the use of a 2D Fast Fourier Transform (FFT) oracle $\bar{\mathcal{F}}_{2D}$. The complexity of computing the 2D FFT on an $H_{\text{in}} \times W_{\text{in}}$ input is $T_{\mathcal{F},2D} = \Theta(D_{\text{in}} \log D_{\text{in}})$.

**Runtime Complexity.** The algorithm iterates over all $n_{\text{in}} \times n_{\text{out}}$ pairs of input and output channels. We analyze the cost of the operations within the loop (Lines 4-7).

1. **Line 4:** $\hat{x}^i \leftarrow \mathcal{F}_{2D} \circ \text{rev}_{2D}(x^i)$. The spatial reversal $\text{rev}_{2D}$ takes $O(D_{\text{in}})$ FLOPS. The 2D FFT takes $T_{\mathcal{F},2D}$ FLOPS.
2. **Line 5:** $\hat{g}^j \leftarrow \mathcal{F}_{2D} R_{2D}^* g^j$. The dilation operation $R_{2D}^*$ (zero-insertion) takes $O(H_{\text{out}} W_{\text{out}})$ FLOPS. The 2D FFT takes $T_{\mathcal{F},2D}$ FLOPS.
3. **Line 6:** $v^{i,j} \leftarrow Q_{2D}^* \circ \text{rev}_{2D} \circ \mathcal{F}_{2D}^{-1}(\hat{x}^i \odot \hat{g}^j)$. The Hadamard product takes $O(D_{\text{in}})$ FLOPS. The inverse 2D FFT ($\mathcal{F}_{2D}^{-1}$) also takes $T_{\mathcal{F},2D}$ FLOPS. The reversal $\text{rev}_{2D}$ takes $O(D_{\text{in}})$ FLOPS. The cropping operation $Q_{2D}^*$ takes $O(H_k W_k)$ FLOPS.
4. **Line 7:** $r^{i,j} \leftarrow \|v^{i,j}\|^2$. Computing the squared norm takes $O(H_k W_k)$ FLOPS.

The total cost per iteration is the sum of these operations:

$$\text{Cost}_{\text{iter}} = 3T_{\mathcal{F},2D} + O(D_{\text{in}} + H_{\text{out}} W_{\text{out}} + H_k W_k).$$

Since $T_{\mathcal{F},2D} = \Theta(D_{\text{in}} \log D_{\text{in}})$, and $H_{\text{out}} W_{\text{out}}, H_k W_k \leq O(D_{\text{in}})$, the dominant cost is driven by the FFT operations. Thus, $\text{Cost}_{\text{iter}} = \Theta(D_{\text{in}} \log D_{\text{in}})$.

The total runtime complexity $T_{\text{fft},2D}$ is the cost per iteration multiplied by the total number of iterations ($n_{\text{in}} n_{\text{out}}$):

$$T_{\text{fft},2D} = n_{\text{in}} n_{\text{out}} \cdot \Theta(D_{\text{in}} \log D_{\text{in}}) = \Theta(n_{\text{in}} n_{\text{out}} H_{\text{in}} W_{\text{in}} \log(H_{\text{in}} W_{\text{in}})).$$

**Storage Complexity.** The algorithm requires storage for the intermediate arrays, such as the Fourier transformed inputs $\hat{x}^i$ and gradients $\hat{g}^j$. These arrays are complex-valued and have sizes proportional to the input dimensions $D_{\text{in}}$. Therefore, the additional storage complexity required is $\Theta(D_{\text{in}})$. $\quad\square$

