# OpenReview forum: "Efficient Gradient Clipping Methods in DP-SGD for Convolution Models"
_ICLR.cc/2026/Conference — Submitted to ICLR 2026_

### Official Review · Reviewer_JyQv · 2025-10-28

**Soundness:** 3
**Presentation:** 2
**Contribution:** 3
**Rating:** 2
**Confidence:** 4

**Summary:**

This paper addresses a known bottleneck in Differentially Private Stochastic Gradient Descent (DP-SGD) — the high computational and memory cost of per-example gradient clipping — particularly in convolutional neural networks (CNNs).
The authors propose three practical methods for improving clipping efficiency. A meta-algorithm dynamically selects among these three methods depending on the model structure and hyperparameters. Benchmark experiments show improved runtime and memory usage compared with existing implementations.

**Strengths:**

- Clear identification of a real bottleneck in DP-SGD training for convolutional models, with quantitative improvements in time and memory complexity.
- Technical soundness: The use of circulant matrix properties and FFTs to accelerate convolution gradient norms is mathematically grounded and novel in the context of DP-SGD clipping.
- Empirical evaluation: The paper provides benchmarks that show consistent efficiency gains over established baselines, demonstrating scalability to larger kernel sizes and batch dimensions.
- Simplicity of integration: The proposed methods could be incorporated into existing DP frameworks such as Opacus without major architectural changes.

**Weaknesses:**

- W1: Lack of experimental depth and reproducibility: The benchmarks are limited to synthetic or minimal CNN settings, with no mention of dataset details or task relevance. As a result, it is unclear how the methods perform in real-world training scenarios (e.g., CIFAR, ImageNet).
- W2: No clear analysis of privacy–utility trade-offs: While efficiency is emphasized, there is no evaluation of whether faster clipping alters gradient accuracy or privacy accounting in practice.
- W3: Missing theoretical or experimental justification for the meta-algorithm’s dispatch policy: The rules governing when each clipping method is chosen remain underexplained and appear heuristic.
- W4: Presentation and structure issues: The paper lacks a formal conclusion or discussion section, and the evaluation figures/tables are not clearly tied to claims in the introduction. As such, the work reads more like a technical note than a complete research paper.

**Questions:**

Address W1-4

---

### Official Review · Reviewer_Mb2h · 2025-11-02

**Soundness:** 3
**Presentation:** 3
**Contribution:** 2
**Rating:** 4
**Confidence:** 4

**Summary:**

This paper studies efficient methods to compute the per-sample gradient norm for DP-SGD and for the convolution layers. Three methods are proposed to improve the inefficiency of DP-SGD, where the DFT method is highlighted and experimented.

**Strengths:**

This paper tackles a sub-problem (convolution layers) of an important problem (DP inefficiency) with a relatively new angle. The DFT method has good scalability in the high-dimensional setting and the efficiency improvement is backed up by experiments on 1D convolution with large kernel. E.g. in Table 4.3, it improves by 2 times over Opacus' implementation.

**Weaknesses:**

1. I am concerned that the experiments (not the method itself) is significantly limited to 1D convolution and one-layer CNN. This deviates away real CNN like ResNet50. Why there are no experiments on multi-layer CNNs with 2D convolution? Can the authors specify what is the usage of multi-layer CNN with 1D convolution?

2. There are 4 methods being experimented in this work: Algorithm 3.3, Lee and Kifer, Bu, and Naive (opacus). But none of the tables compare all four of them. Is there a technical difficulty to do so? In particular, in Table 4.3, the comparison between Algorithm 3.3 and Opacus is not a strong and useful comparison. It is well-known that opacus is memory hungry and there are more efficient libraries to compute DP grad for convolution (mostly 2D) like mixed ghost clipping. Can the authors add experiments on that?

**Questions:**

See weaknesses.

---

### Official Review · Reviewer_U67n · 2025-11-02

**Soundness:** 3
**Presentation:** 2
**Contribution:** 2
**Rating:** 4
**Confidence:** 3

**Summary:**

This paper proposes FFT or DFT method to compute the per-sample gradient norm for DP-SGD. The method only works for convolution layers theoretically and for 1D convolution layers empirically. Theoretical analysis on complexity has shown improvement in high dimension. Empirical comparison has shown improvement over memory-heavy naive implementation.

**Strengths:**

Implementation inefficiency of DP is critical in practice and solving it with FFT is original. The derivation and complexity analysis is sound as far as I am concerned. In the specific scenario of 1D convolution, the improvement is of certain significance. The paper is clear in stating its applicability when kernel size is large.

**Weaknesses:**

While I understand the derivation in 1D and 2D convolution, the empirical experiments seem incomplete. Is there only one-layer model results? In my opinion, multi-layer 2D CNN is the norm and even this may be challenged by vision transformers that do not necessarily use convolution.

Suppose this method works with large-kernel 2D CNN, I think the baselines are abundant: Opacus, fastDP, private_vision, private_transformers, etc. Currently this work seems to focus on the theoretical innovation without touching on the algorithmic development, right? If we move to larger scale, do the authros expect benefit or blockers in distributed DP training?

**Questions:**

NA

---

### Official Review · Reviewer_ot86 · 2025-11-02

**Soundness:** 3
**Presentation:** 3
**Contribution:** 2
**Rating:** 4
**Confidence:** 3

**Summary:**

The paper proposes a few ways to perform per-sample gradient clipping for convolution kernels efficiently, which is useful for DP training algorithms on models using convolution. Results show the proposed algorithms are faster than standard gradient clipping in existing libraries when kernel size is relatively large.

**Strengths:**

1. Efficient gradient clipping is import for differetially-private training algorithms. The efficient gradient clipping algorithms on convolution kernels will be useful in practice.

**Weaknesses:**

1. Experiments only on 1d kernels and there is no open-source code. The derivation of the proposed methods are straightforward from gradient calculation rules. I would expect more practical contribution from the paper.

**Questions:**

Why only 1d kernels are tested in experiments and do authors have plan to opensource code?

---

### Meta-Review · Area_Chair_XSWr · 2025-12-17

**Summary:**

This paper presents a novel and mathematically sound method using FFT to accelerate gradient clipping in DP-SGD for convolutional layers, addressing a recognized efficiency bottleneck. Reviewers agree the core idea is original and the complexity analysis is compelling. However, the work is currently seen as a promising technical note rather than a complete contribution, due to significant experimental limitations. The evaluation is restricted to synthetic settings and 1D convolutions, lacking validation on standard multi-layer 2D CNNs (e.g., ResNet) with real-world datasets and comparisons to modern DP libraries. Critical aspects like the privacy-utility trade-off, integration into distributed training, and the heuristic dispatch policy are unexamined. To demonstrate practical impact, I think the major revisions are required, such as extending experiments to mainstream 2D vision tasks and stronger baselines, providing an open-source implementation, and thoroughly analyzing the method's effect on final model utility and scalability.

**Reviewer Concerns:**

There is no rebuttal from the authors such that all concerns are still outstanding.

**Reviewer Scores:**

There is no discussion between the reviewers and the authors such that the reviewer scores remain the same.

---

### Decision · Program_Chairs · 2026-01-26

Reject